The rice XA21 ectodomain fused to the Arabidopsis EFR cytoplasmic domain confers resistance to Xanthomonas oryzae pv. oryzae

http://orcid.org/0000-0002-3241-975X Thomas Nicholas C. 1 2
Oksenberg Nir 1
Liu Furong 1
Caddell Daniel 1
Nalyvayko Alina 1
Nguyen Yen 1
Schwessinger Benjamin 1 3 benjamin.schwessinger@anu.edu.au
Ronald Pamela C. 1 pcronald@ucdavis.edu
1 Department of Plant Pathology and the Genome Center, University of California, Davis , Davis, CA , USA
McCormick Sheila
2 Current affiliation: Department of Plant Pathology and Microbiology, University of California, Riverside, Riverside, CA, USA

3 Current affiliation: Research School of Biology, Australian National University, Acton, MA, Australia

Electronic publication date: 2018 May 9
Publication date: 2018
Volume: 6
Electronic Location ID: e4456
Received 2017 Apr 19; Accepted 2018 Feb 15
Copyright: © 2018 Thomas et al.
Copyright year: 2018
Copyright holder: Thomas et al.
License: This is an open access article distributed under the terms of the Creative Commons Attribution License, which permits unrestricted use, distribution, reproduction and adaptation in any medium and for any purpose provided that it is properly attributed. For attribution, the original author(s), title, publication source (PeerJ) and either DOI or URL of the article must be cited.
License URL: https://creativecommons.org/licenses/by/4.0/

Keywords: Rice, XA21, EFR, Disease resistance, Xanthomonas, Immune receptor, Chimeric receptor

Funding: NIH grant GM59962 NSF PGRP grant IOS-1237975 Human Frontiers Science Program long-term postdoctoral fellowship LT000674/2012 Discovery Early Career Research Award DE150101897 This project was funded through NIH grant GM59962 and the NSF PGRP grant IOS-1237975. Benjamin Schwessinger was supported by a Human Frontiers Science Program long-term postdoctoral fellowship (LT000674/2012) and a Discovery Early Career Research Award (DE150101897). The funders had no role in study design, data collection and analysis, decision to publish, or preparation of the manuscript.

==============================
Rice (Oryza sativa) plants expressing the XA21 cell-surface receptor kinase are resistant to Xanthomonas oryzae pv. oryzae (Xoo) infection. We previously demonstrated that expressing a chimeric protein containing the ELONGATION FACTOR Tu RECEPTOR (EFR) ectodomain and the XA21 endodomain (EFR:XA21) in rice does not confer robust resistance to Xoo. To test if the XA21 ectodomain is required for Xoo resistance, we produced transgenic rice lines expressing a chimeric protein consisting of the XA21 ectodomain and EFR endodomain (XA21:EFR) and inoculated these lines with Xoo. We also tested if the XA21:EFR rice plants respond to a synthetic sulfated 21 amino acid derivative (RaxX21-sY) of the activator of XA21-mediated immunity, RaxX. We found that five independently transformed XA21:EFR rice lines displayed resistance to Xoo as measured by lesion length analysis, and showed that five lines share characteristic markers of the XA21 defense response (generation of reactive oxygen species and defense response gene expression) after treatment with RaxX21-sY. Our results indicate that expression of the XA21:EFR chimeric receptor in rice confers resistance to Xoo. These results suggest that the endodomain of the EFR and XA21 immune receptors are interchangeable and the XA21 ectodomain is the key determinant conferring robust resistance to Xoo.

Introduction

Plant cell-surface immune receptors confer defense against pathogen infection. Cell-surface mediated immunity in plants is mainly conferred by receptor-like proteins (RLPs) and receptor-like kinases (RLKs) that recognize pathogen associated molecular patterns (Jones & Dangl, 2006; Macho & Zipfel, 2014). Three well-studied cell-surface RLKs that confer resistance to bacterial pathogens include FLAGELLIN SENSING2 (FLS2; At5G6330) (Gómez-Gómez & Boller, 2000), EF-TU RECEPTOR (EFR; At5g20480) (Zipfel et al., 2006) from Arabidopsis and XA21 (U37133) from Oryza longistaminata (Song et al., 1995). The identification of the microbial molecules recognized by these three receptors have enhanced in depth characterization of their functional properties. The FLS2 receptor binds the flg22 peptide derived from bacterial flagellin (Felix et al., 1999; Gómez-Gómez & Boller, 2000; Chinchilla et al., 2006). EFR recognizes the elf18 peptide derived from the bacterial Elongation Factor Thermo-unstable protein (EF-Tu) (Kunze et al., 2004; Zipfel et al., 2006). XA21 recognizes the sulfated required for activation of Xa21-mediated immunity X (RaxX) protein produced by Xanthomonas oryzae pv. oryzae (Xoo; Pruitt et al., 2015). Although these receptors specifically recognize different molecules, they share similar domain structures including ectodomains containing leucine rich repeats and endodomains containing intracellular kinases of the non-arginine aspartate (non-RD) class (Dardick & Ronald, 2006).

Domain swap studies between cell-surface receptors have led to the hypothesis that the nature of the endodomains is the primary determinant dictating the specific disease resistance outcome. For example, studies of a chimeric receptor generated by fusion of the Arabidopsis BRASSINOSTEROID-INSENSITIVE1 (BRI1) receptor, which recognizes brassinosteroid hormones, to the XA21 endodomain (BRI1:XA21) (Li & Chory, 1997) indicated that the chimeric receptor could be activated by brassinosteroid treatment. Rice cells expressing BRI1:XA21 (NRG-1 in the original publication) and treated with brassinosteroid initiated cell death, produced reactive oxygen species (ROS), and expressed stress-related genes. The stress-related symptoms were attributed to the activation of the XA21 endodomain because the full-length BRI1 receptor does not induce the same stress-related symptoms as BRI1:XA21 (He et al., 2000). These results suggested that the XA21 endodomain was activated upon BRI1 recognition of brassinosteroid and that the specific type of response was most consistent with the response mediated by the XA21 endodomain and not the BRI1 ectodomain.

Another chimera study compared the responses of receptors consisting of ectodomain and endodomain exchanges between EFR and WAK1. WAK1 recognizes oligogalacturonides (OGs) released from damaged plant cell walls (Decreux & Messiaen, 2005; Decreux et al., 2006; Cabrera et al., 2008). Elf18 treated wild-type plants and OG treated plants expressing a WAK1:EFR (WEG) chimeric RLK both produced ROS, ethylene and expressed the EFR-induced genes (At3g22270 and At4g37640) while EFR:WAK1 (EWAK) expressing plants did not. Instead, EWAK plants retained WAK1-like responses by producing ROS but not ethylene in response to OGs (Ferrari et al., 2008; Brutus et al., 2010). WEG and EWAK responses were therefore most consistent with the response conferred by the respective endodomain portion of each fusion protein. Another study showed that fusing the XA21 endodomain to the fungal chitin RLP CEBiP (Kaku et al., 2006) (CRXA-1, and CRXA-3 in the original publication) conferred a more robust immune response to fungal infection by Magnaporthe oryzae than when expressing or overexpressing CEBiP alone (Kishimoto et al., 2010). These results suggested that the XA21 endodomain was responsible for conferring the enhanced immune response to M. oryzae. Together, these studies indicate that the endodomain of several immune receptors dictate the specific signaling events that lead to disease resistance in whole plants or defense responses in plant cells. These studies also suggest chimeras carrying the XA21 endodomain, when treated with the appropriate ligand, can initiate an immune response similar to that mediated by the full-length receptor XA21.

To further explore the function and specificity of the XA21 endodomain and ectodomain, we previously generated transgenic rice lines expressing EFR, tagged with green fluorescent protein (EFR:GFP), or a chimeric EFR:XA21 protein, consisting of the EFR ectodomain and the XA21 transmembrane and intracellular domain, tagged with GFP (EFR:XA21:GFP). The maize ubiquitin promoter drove expression for both EFR:GFP and EFR:XA21:GFP transgenes (Schwessinger et al., 2015a). Both EFR:GFP and EFR:XA21:GFP rice plants were susceptible to Xoo strain PXO99A and conferred partial resistance to weakly virulent strains, which suggested EF-Tu from Xoo was still recognized by EFR:GFP rice (Schwessinger et al., 2015a). These studies suggested that although both receptors were capable of recognizing EF-Tu, they were still unable to initiate a robust immune response to PXO99A. As noted in the paper’s discussion, these results were counterintuitive based on earlier domain swap studies that indicated that the endodomain dictates immune signaling and disease resistance (He et al., 2000; Brutus et al., 2010; Albert et al., 2010; Kishimoto et al., 2010).

Although it is unclear why the EFR and EFR:XA21 study conflicted with findings from previous chimeric receptor studies, there are several possibilities to explain these discrepancies. In the case of the EFR:WAK1 and WAK1:EFR study, it could be that the type of kinase domain dictated the distinct signaling mediated by each chimeric receptor because the WAK1 and EFR kinase domains belong to different kinase classes. The WAK1 kinase domain contains an arginine (R) aspartate (D) motif while the EFR kinase domain is non-RD, as described above. The non-RD kinases are almost always associated with immune responses in plants and animals and are likely regulated differently than RD kinases (Dardick & Ronald, 2006; Ronald & Beutler, 2010; Dardick, Schwessinger & Ronald, 2012). Thus, the presence of the non-RD domain may dictate an immune response when appropriately activated and the presence of the RD domain may specify a WAK1-like response.

For both BRI1:XA21 and CEBiP:XA21 studies, it is possible that the origin of the kinase domain from XA21 was less important than the fact that the kinase belonged to the non-RD class. For example, it is unclear if fusing BRI1 or CEBiP to other non-RD kinases, such as the kinases from EFR or OsFLS2, would have produced similar results (Takai et al., 2008).

Previous studies have shown that the XA21 ectodomain plays a critical role in the immune response. For example, the Xa21D paralog, which lacks a transmembrane and intracellular domain, confer partial resistance to Xoo (Wang et al., 1998). Unlike Xa21, Xa21D only encodes an ectodomain that is nearly identical to the XA21 ectodomain, differing only in 15 amino acid residues compared to the XA21 ectodomain. Similarly, expression of a catalytically inactive variant of XA21, carrying a mutation in the catalytic domain of the kinase (K736E), in rice maintained partial resistance to Xoo (Andaya & Ronald, 2003). Together, these studies indicate that the XA21 ectodomain is sufficient to confer partial resistance to Xoo, even in the absence of a functional kinase domain.

To further explore the function and importance of the XA21 ectodomain, we generated transgenic rice lines expressing a chimeric protein containing the XA21 ectodomain fused to the EFR transmembrane and intracellular domain, tagged with GFP (XA21:EFR:GFP) (Holton et al., 2015). We found that XA21:EFR:GFP rice display robust resistance to Xoo strain PXO99A. We also show that XA21:EFR:GFP was specifically activated by RaxX as measured by bacterial infection, defense response gene expression and ROS production (Pruitt et al., 2015; Schwessinger et al., 2015b; Wei et al., 2016). These results indicate that the XA21 ectodomain and its recognition of RaxX specify robust resistance to Xoo even in the absence of the XA21 endodomain.

Materials and Methods

Plant material and methods

Rice seeds were germinated on water-soaked filter paper for 5–7 days at 28 °C and then transplanted into 2.6-l pots. Plants were grown in an approximately 80/20 (sand/peat) soil mixture in an environmentally-controlled greenhouse with temperature set between 28 and 30 °C with 75–80% humidity.

Transgenic rice production

The Xa21:EFR:GFP (XA21 aa residues 1–650 fused to EFR aa residues 650–1,031) binary vector used in rice transformation was described previously (Holton et al., 2015). Transgenic Kitaake plants expressing the Xa21:EFR:GFP transgene, under control of the maize ubiquitin promoter, were generated by the UC Davis Plant Transformation Facility as described previously (Hiei et al., 1994). pCAMBIA1300 binary vectors carrying the Xa21:EFR:GFP construct were transformed into Kitaake calli by Agrobacterium-mediated transformation. Regenerated plants were selected on hygromycin. The presence of the transgene was confirmed in each generation by PCR using transgene-specific primers (Table S1).

Bacterial infection of rice plants

Xoo isolates PXO99A, PXO99AΔraxX, and PXO99AΔraxX(raxX) (Pruitt et al., 2015) were plated on peptone sucrose agar plates for three days. Xoo was suspended in water to approximately 5 × 108 colony forming units (CFU)/mL for inoculation. Greenhouse-grown plants were transported into environmentally-controlled growth chambers at the four-week-old stage. Chamber conditions were set to 26 °C, 85% humidity with 12 h light/dark cycles. Plants were acclimated to the chamber conditions for two to three days before scissor inoculation (Kauffman et al., 1973).

Segregation analysis

The presence of each transgene was identified using PCR genotyping using genomic DNA templates and transgene-specific primers (Table S1). Chi-square tests were used to determine possible multiple transgene insertions.

Gene expression analysis by qRT-PCR

Total RNA was extracted from detached leaves frozen in liquid nitrogen and powdered using a Qiagen tissuelyser. RNA was extracted from powdered tissue using TRI Reagent and precipitated with isopropanol. RNA was DNase treated using the TURBO DNase kit from Life Technologies, Carlsbad, CA USA. RNA concentrations were normalized to the lowest sample concentration in each experiment. cDNA was synthesized from 2 μg of total RNA using the High Capacity cDNA Reverse Transcription Kit by Life Technologies, Carlsbad, CA USA. Gene expression changes were determined by ΔΔCt method (Livak & Schmittgen, 2001) normalizing gene expression to Actin (LOC_Os03g50885) and using mock treated samples as the reference for stress gene expression. Quantitative real time PCR (qRT-PCR) was performed using a Bio-Rad CFX96 Real-Time System coupled to a C1000 Thermal Cycler (Bio-Rad, Hercules, CA USA) using the Bio-Rad SsoFast EvaGreen Supermix. qRT-PCR primer pairs used are described in Table S1. qRT-PCR reactions were run for 40 cycles with annealing and amplification at 62 °C for 5 s and denaturation at 95 °C for 5 s. Single melting curves were observed for all primer pairs used indicating negligible off-target amplification.

Western blot analysis for protein expression

Anti-GFP (Santa Cruz Biotech, Santa Cruz, CA USA) was used to detect EFR:GFP, EFR:XA21:GFP, XA21:GFP, and XA21:EFR:GFP. Secondary anti-mouse antibodies (Santa Cruz Biotech) conjugated to horseradish peroxidase were used in combination with chemiluminescence substrates (Thermo, Waltham, MA USA) to detect proteins on a Biorad ChemiDoc.

Reactive oxygen species production

Leaves of three- to four-week-old rice plants were cut longitudinally along the mid vein and then into 1–1.5 mm thick pieces. Leaf pieces were floated on sterile water overnight. The following morning, two leaf pieces were transferred into one well of a 96-well white plate containing 100 μl elicitation solution (20 μM L-012 (Wako Chemicals, Neuss, Germany), 2 μg/mL HRP (Sigma, St. Louis, MO, USA)). A total of 500 nM of elf18 (Escherichia coli) or RaxX21-sY peptides were used for treatments. ROS production was measured for 0.5 s per reading with a high sensitivity plate reader (TriStar; Berthold, Bad Wildbad, Germany).

Results

Transgenic rice expressing the XA21:EFR chimeric receptor display robust resistance to Xoo

We produced transgenic rice lines expressing an Xa21:EFR:GFP chimeric construct to test whether the XA21 ectodomain confers resistance to Xoo when fused to the EFR cytoplasmic domain. This construct encodes the XA21 ectodomain (XA21 residues 1–650) fused to the EFR transmembrane, juxtamembrane, and cytoplasmic domain (EFR residues 651–1,031) with a carboxyl-terminal GFP fusion (Holton et al., 2015) expressed under the maize ubiquitin promoter. We generated 10 independent transgenic T0 lines and inoculated the plants with Xoo using a leaf clipping method followed by lesion length measurements, which allows a comparable but more rapid assessment of resistance than bacterial population counting. We found that eight of these lines (lines 2, 3, 4, 5, 6, 7, 9, and 10) displayed enhanced resistance to Xoo compared with the Kitaake parent line (Fig. S1).

To assess if the resistance phenotype was transmitted to the next generation, we self-pollinated five of the eight T0 lines (lines 2, 4, 5, 6, 7) and collected T1 seed. These T1 plants, as well as rice plants expressing and lacking Xa21 as controls, were inoculated with Xoo and assessed for resistance by measuring the lengths of disease-induced lesions. We observed that T1 individuals that were PCR positive for the transgene in lines 2, 4, 5, and 6 co-segregated with resistance to Xoo (PCR positive to negative ratios 8:4, 21:0, 8:7, and 16:5, respectively). Lesion length averages were approximately 5 cm in resistant individuals compared to approximately 13 cm for susceptible controls (Fig. 1 and Fig. S2). All T1 individuals from line 4 were PCR positive for Xa21:EFR:GFP (21:0) which could have been from multiple transgene insertions (X2 (1) = 1.4, p = 0.24) and were resistant to Xoo. All T1 individuals from line 7 were also PCR positive for the Xa21:EFR:GFP transgene. However, these plants showed varying degrees of resistance (Fig. S2).

Figure 1 Rice expressing Xa21:EFR:GFP are resistant to Xoo infection.

(A) The bar graph represents the average lesion length observed on rice plants infected with Xoo. Control lines used were Kitaake, EFR:GFP, EFR:XA21:GFP, and Myc:XA21 rice (green bars). Experimental samples include individuals PCR positive for the Xa21:EFR:GFP transgene (black bars) from line 2 and 6 and PCR negative individuals (red bars). Five-week-old greenhouse-grown plants were scissor inoculated with PXO99A (5 × 108 CFU/mL) and disease lesions were scored approximately two weeks post inoculation. Error bars represent standard deviation from the mean lesion length. Mean lesion lengths are the average of lesion measurements from individual leaves from the same plant (n ≥ 3). Black lines and letters above the graph represent statistical groupings using the Tukey–Kramer HSD test. Different letters indicate significant differences (p < 0.05). This experiment was repeated at least three times with similar results. (B) Photograph of select leaves from the same experiment in A. The photograph shows Kitaake, EFR:GFP, EFR:Xa21:GFP, Myc:Xa21, Xa21:EFR:GFP individual -2-23, and -6-5-17 leaves infected with Xoo and was taken approximately two weeks after inoculation. Nicholas Thomas provided the photograph.

To determine if Xoo resistance in Xa21:EFR:GFP plants is mediated through XA21 perception of RaxX, we infected two different T1 progeny from two different -2 lines (-2-13 and -2-19) and T2 progeny from line -6-5-1 with wild-type PXO99A, PXO99A ΔraxX mutants (ΔraxX) that evade full-length XA21-medaited immunity, and ΔraxX strains complemented with raxX (ΔraxX(raxX)) that do no evade XA21-mediated immunity (Pruitt et al., 2015). We found that Kitaake plants were susceptible to all strains used (average lesion lengths approximately 20 cm), whereas Xa21:GFP control plants were susceptible to ΔraxX infections with approximately 19 cm average lesion lengths and resistant to WT PXO99A and complemented ΔraxX(raxX) strains with approximately 7 cm average lesion lengths consistent with previous findings (Pruitt et al., 2015). For segregants carrying the Xa21:EFR:GFP transgene, plants were significantly more resistant to WT PXO99A, with approximately 5 cm average lesion lengths, compared to ΔraxX infections that developed lesions approximately 17 cm long. Xa21:EFR:GFP segregants carrying the transgene were significantly more resistant to ΔraxX(raxX) with approximately 8 cm average lesion lengths compared to null segregants that developed significantly longer average lesion lengths of approximately 17 cm (Fig. S3). These results indicate that the XA21:EFR:GFP protein perceives RaxX.

For subsequent experiments, we focused on two Xa21:EFR:GFP lines (-2 and -6) for further molecular characterization. For these experiments, T1 plants were used for line 2 and T2 plants were used for line 6 (we self-pollinated T1 individuals and collected T2 seed for line 6) to test if similar phenotypes are observable in different lines and in subsequent generations. We found that T2 individuals from line 6 maintained Xoo resistance that segregated with the Xa21:EFR:GFP transgene (Fig. 1). Because T1 and T2 individuals from lines 2 and 6, respectively, were still segregating for the transgene, we performed experiments on individual plants that carried the Xa21:EFR:GFP transgene, selected by PCR genotyping. We used null segregant individuals as controls.

The Xa21:EFR:GFP chimeric transgene is expressed and XA21:EFR:GFP protein accumulates in stable transgenic lines

We used qRT-PCR to assess if plants containing the Xa21:EFR:GFP transgene express the Xa21 ectodomain and EFR cytoplasmic domain. We assessed transcript levels using domain-specific primers for regions that encode the XA21 ectodomain, XA21 cytoplasmic domain, and the EFR cytoplasmic domain (Table S1) (Figs. 2A–2C). Our results show Xa21:EFR:GFP-2-23, -2-24, -6-5-17, and -6-5-18 that carry the transgene specifically express regions encoding the XA21 ectodomain and the EFR cytoplasmic domain. Additionally, these plants do not express regions encoding the XA21 endodomain. Because these plants are not expressing the full-length Xa21 transcript or endodomain, any immune responses observed in these plants are not mediated by full-length XA21 or the XA21 endodomain.

Figure 2 Xa21:EFR:GFP transcripts are expressed in stable transgenic lines.

Bar graphs represent the relative expression of transgenic transcripts. (A) Relative amplification of the Xa21 endodomain with Myc:Xa21 rice as the expression reference. (B) Amplification of the Xa21 endodomain with Myc:Xa21 rice as the expression reference. (C) Amplification of the EFR cytoplasmic domain with EFR:GFP rice as the expression reference. Gene expression was measured by quantitative real-time PCR using cDNA amplified from total RNA as a template. Each gene expression measurement is the average of two technical replicates and error bars represent the standard deviation between the two measurements.

In addition to the specific Xa21:EFR:GFP transcript, we show that XA21:EFR:GFP protein accumulates in transgenic rice. We performed Western Blot analysis to determine if XA21:EFR:GFP protein accumulates in Xa21:EFR:GFP transgenic rice using primary anti-GFP antibodies. Our results show that XA21:EFR:GFP protein is detectable in Xa21:EFR:GFP-2-28, -2-29, -6-5-4, and -6-5-7 that carry the Xa21:EFR:GFP transgene. Wild-type Kitaake and null segregants Xa21:EFR:GFP-2-32 and Xa21:EFR:GFP-6-5-6 do not express any GFP tagged protein (Fig. S3). Together, RNA and protein expression indicate that two independent Xa21:EFR:GFP transgenic lines express Xa21:EFR:GFP transcript and accumulate XA21:EFR:GFP protein.

RaxX21-sY treated Xa21:EFR:GFP rice leaves produce reactive oxygen species and highly express stress-related genes

We next assessed if Xa21:EFR:GFP rice are able to activate immune responses after RaxX treatments. We used a commercially synthesized, sulfated RaxX peptide, composed of 21 amino acids from the Xoo RaxX protein sequence in PXO99A (RaxX21-sY) previously shown to activate XA21-mediated immunity (Pruitt et al., 2015; Wei et al., 2016). Bursts of ROS are commonly measured to assess immune responses because ROS are rapidly produced as a defense response to pathogen attack (Wojtaszek, 1997; Jones & Dangl, 2006; Macho & Zipfel, 2014). We therefore measured ROS production in Xa21:EFR:GFP rice after RaxX21-sY treatment to determine if plants carrying the chimeric protein respond similarly to RaxX21-sY treated plants carrying full-length XA21 (Pruitt et al., 2015). Xa21:EFR:GFP rice accumulate ROS in response to RaxX21-sY treatments, but not to mock or elf18 treatments (Fig. 3). In addition, we confirmed that RaxX21-sY treated XA21:GFP rice, expressing the full-length XA21 protein tagged with GFP (Fig. S4), accumulate ROS (Fig. 3B). Null segregants did not produce ROS bursts in response to RaxX21-sY treatments (Fig. 3). EFR:GFP and EFR:XA21:GFP rice responded to elf18, but not to RaxX21-sY, showing that the XA21 ectodomain in full-length XA21 and XA21:EFR:GFP proteins is necessary for RaxX-triggered immune responses (Figs. 3C and 3D).

Figure 3 Reactive oxygen species accumulate after peptide treatments.

Reactive oxygen species (ROS) production after water (mock, blue diamonds), 500 nM RaxX21-sY (red squares), or 500 nM elf18 peptide treatments (green triangles). (A) ROS production in wild-type Kitaake rice and (B) Xa21:GFP rice, (C) EFR:GFP rice and (D) EFR:Xa21:GFP rice. (E) and (G) ROS production in T1 and T2 null-segregant individuals from Xa21:EFR:GFP line -2 and line -6-5, respectively. (F) and (H) ROS production in T1 and T2 ndividuals from line -2 and -6-5, respectively, that segregate for the Xa21:EFR:GFP transgene. Each datapoint represents an average of four technical replicate measurements and error bars represent the standard error of the averages. These experiments have been repeated three times with similar results.

We next measured stress-related marker gene expression in RaxX21-sY treated Xa21:EFR:GFP rice to further characterize the XA21:EFR:GFP-mediated response. We measured the expression of rice defense marker genes PR10b, LOC_Os02g36190, LOC_Os06g37224, and LOC_Os11g42200. PR10b encodes a putative ribonuclease and is up-regulated upon fungal infection in rice (McGee, Hamer & Hodges, 2001), LOC_Os02g36190 is involved in phytoalexin biosynthesis and bacterial blight resistance (Li et al., 2013), LOC_Os06g37224 encodes an ent-Kaurene Oxidase up-regulated after ultraviolet light stress (Itoh et al., 2004), and LOC_Os11g42200 encodes a laccase precursor protein and is up-regulated after Atrazine herbicide treatment (Huang et al., 2016). The up-regulation of these genes were previously established as markers of the rice stress-response including the XA21-mediated immune response to sulfated RaxX peptides and to Xoo infection (Chen et al., 2014; Pruitt et al., 2015; Thomas et al., 2016) using a detached leaf treatment assay. RNA was extracted from detached leaves of four-week-old plants mock treated with water or with 500 nM of RaxX21-sY for 6 h. Gene expression was measured in individuals Xa21:EFR:GFP-2-23 and -6-5-17 by quantitative real-time PCR. Higher expression was observed in each of the stress-related genetic markers in RaxX21-sY treated Myc:Xa21 and Xa21:EFR:GFP rice (Figs. 4A–4D). Gene expression was not induced in any mock treated samples or Kitaake samples treated with RaxX21-sY. Only Individual 2–23 showed higher induction of LOC_Os11g42200 and LOC_Os06g37224, and only induction of LOC_Os06g37224 was significant (Fig. 4C). Although the magnitude of ROS and stress gene induction in response to RaxX21-sY is relatively lower in Xa21:EFR:GFP rice compared to XA21:GFP rice (Figs. 3 and 4), both show robust resistance to Xoo infection as measured by lesion length progression (Fig. 1) Together, the results from ROS and gene expression experiments suggest that the XA21 ectodomain in XA21:EFR:GFP is sufficient to recognize RaxX and that the EFR endodomain can be substituted for the XA21 endodomain to transduce immune responses after RaxX treatment.

Figure 4 Xa21:EFR:GFP rice express stress-related genes after RaxX21-sY treatment.

Gene expression profiles of four stress-related genes, (A) PR10b, (B) LOC_Os2g36190, (C) LOC_06g37224, and (D) LOC_Os11g42200. Samples are rice leaves from wild-type Kitaake, Myc:XA21 rice, and individuals -23 from Xa21:EFR:GFP line -2 and individual -17 from line -6-5. Leaves were mock treated with water (-) or with 500 nM RaxX21-sY for Myc:XA21 and XA21:EFR rice and 500 nM elf18 for EFR:GFP and EFR:XA21:GFP rice (+). Kitaake was treated with both RaxX21-sY and elf18 in these experiments. Letters indicate significant difference in gene expression compared to mock using the Tukey–Kramer HSD test (α = 0.05). Expression levels are normalized to mock treatment of the same line. Bars depict average expression level relative to actin expression ± standard error of three technical replicates. This experiment was repeated twice with similar results.

Discussion

Here we show that the ectodomain of XA21 is sufficient to confer full resistance to Xoo strain PXO99A when fused to the intracellular domain of the Arabidopsis immune receptor EFR (Fig. 1 and Figs. S1–S3). We previously demonstrated that a functional EFR:XA21:GFP is not able to confer resistance to Xoo when expressed in rice. Together these results suggest that the XA21 extracellular domain and the recognition of RaxX are the key properties that dictate the robust immune response of XA21. Both the native XA21 endodomain as well as the EFR endodomain fused to the XA21 ectodomain appear to be interchangeable as both XA21 and EFR kinases can confer robust resistance when fused with the XA21 ectodomain. This result slightly contrasts with previous domain swap studies that indicated that the endodomains of immune receptors were the defining properties of the immune receptor responses (He et al., 2000; Brutus et al., 2010; Albert et al., 2010; Kishimoto et al., 2010). Although XA21 mutants and XA21 derivatives that lack a functional kinase domain maintain partial resistance, it appears that a functional kinase domain is required for robust resistance (Wang et al., 1998; Andaya & Ronald, 2003).

Despite the evidence that rice expressing XA21:EFR:GFP are resistant to Xoo, it is unclear why plants expressing the reciprocal EFR:XA21:GFP protein are susceptible to Xoo (Schwessinger et al., 2015a). Previous results indicate that elf18 and RaxX21-sY have similar EC50 values for immune activation (Schwessinger et al., 2015a; Pruitt et al., 2015), but it is unclear if RaxX and EF-Tu are available at similar levels during infection, and it is also possible that Xoo masks EF-Tu, preventing optimal EFR ectodomain recognition. We hypothesize that the XA21 ectodomain is critical for conferring robust resistance because it interacts with additional rice-specific signaling components that the EFR ectodomain is unable to bind. In partial support of this hypothesis, we previously showed that the EFR kinase domain does not interact with some of the previously identified XA21 kinase domain signaling components, including the negative regulator XB15 and positive regulator XB3 (Schwessinger et al., 2015a). It is therefore possible that unidentified positive regulators of XA21-mediated immunity that interact with the XA21 ectodomain do not associate with the EFR ectodomain. Future studies might be aimed at identifying these elusive ectodomain-specific signaling partners to better understand XA21-mediated immunity.

Supplemental Information

Supplemental Information 1 Inoculation of Xa21:EFR:GFP T0 lines with Xoo strain PXO99A.

Average lesion lengths from ten independent T0 Xa21:EFR:GFP lines inoculated with Xoo PXO99A. Kitaake and XA21-Kitaake (XA21) rice were used as susceptible and resistant controls (green bars) and inoculated at the 5-week old stage. Transgenic Xa21:EFR:GFP rice were inoculated using the scissor clipping method approximately 4–5 weeks after regeneration. Plants were scored 14 days post inoculation. Error bars represent standard deviation of the mean lesion length measured from multiple leaves from the same plant (n ≥ 3). Statistical analysis was performed using the Tukey-Kramer HSD test for each individual experiment. Different letters indicate significant differences between averages (alpha = 0.05).

Click here for additional data file.

Supplemental Information 2 Inoculation of Xa21:EFR:GFP T1 plants with Xoo strain PXO99A.

T0 progeny from lines 2, 4, 5, 6 and 7 were inoculated with Xoo strain PXO99A. Black bars indicate T1 progeny that carry the Xa21:EFR:GFP transgene while red bars indicate null segregants. Kitaake, EFR:GFP, and EFR:Xa21:GFP rice controls are represented by green bars. +/− represents presence or absence of the Xa21:EFR:GFP transgene determined by PCR. Plants were inoculated at the 5 week-old stage and lesions were scored 14 days post inoculation. Control mean lesion lengths are calculated from pooled lesion measurements (n ≥ 9) from multiple plants and error bars for controls represent the standard deviation. Error bars for experimental samples represent standard deviation of the mean lesion lengths measured from multiple leaves (n ≥ 3) from the same plant. Statistical analysis was performed using the Tukey-Kramer HSD test for each individual experiment. Different letters indicate significant differences between means (alpha = 0.05).

Click here for additional data file.

Supplemental Information 3 Inoculation of Xa21:EFR:GFP T1 and T2 progeny with Xoo ΔraxX strains and complemented ΔraxX(raxX) strains.

T1 progeny from line -2-13, -2-19 and T2 progeny from line -6-5-1 were inoculated with Xoo strains PXO99A, a PXO99A with a deletion in the raxX gene (ΔraxX), and ΔraxX strains complemented with raxX (ΔraxX(raxX)). Plants were inoculated at the 5 week-old stage and lesions were scored 14 days post inoculation. Black bars indicate average lesion length measurements (n > 7) from multiple plants that carry the Xa21:EFR:GFP transgene. Red bars indicate average measurements (n ≥ 3) from single null segregant individuals infected with ΔraxX(raxX). Green bars represent control average lesion length measurements (n ≥ 4) pooled from multiple Kitaake and Xa21:GFP rice plants. Error bars represent standard deviation. Asterisks (*) represent statistically significant differences between the indicated infections using the student’s T-test (p < 0.001). These experiments were repeated twice with similar results.

Click here for additional data file.

Supplemental Information 4 Xa21:EFR:GFP rice containing the transgene express GFP-tagged chimeric protein.

Western blot showing protein level of EFR:GFP, EFR:XA21:GFP, XA21:GFP, and XA21:EFR:GFP using an anti-GFP antibody to detect the C-terminal GFP tag. The lower panel shows the coomassie brilliant blue staining of the membrane as a loading control. + and − indicates the presence of the transgene determined by PCR. XA21:EFR:GFP samples were from T1 individuals XA21:EFR:GFP-2-28, -2-29, and -2-32 and T2 individuals XA21:EFR:GFP-6-5-4, -6-5-6, and -6-5-7.

Click here for additional data file.

Supplemental Information 5 List of primers used in this study.

Table indicates each primer used in this study. The first column shows the 5′ to 3′ primer sequence. The second column indicates the associated MSU locus ID, primer direction, and purpose of primer. The last column indicates figures associated with each primer set.

Click here for additional data file.

We would like to thank Dr. Nicholas Holton and Prof. Dr. Cyril Zipfel from the Sainsbury Laboratory for providing the Xa21:EFR:GFP construct.

Additional Information and Declarations

Competing Interests

Author Contributions

Data Availability

Pamela C. Ronald is an Academic Editor for PeerJ.

Nicholas C. Thomas conceived and designed the experiments, performed the experiments, analyzed the data, prepared figures and/or tables, authored or reviewed drafts of the paper, approved the final draft.

Nir Oksenberg conceived and designed the experiments, performed the experiments, analyzed the data, prepared figures and/or tables, authored or reviewed drafts of the paper, approved the final draft.

Furong Liu performed the experiments, authored or reviewed drafts of the paper, approved the final draft.

Daniel Caddell performed the experiments, authored or reviewed drafts of the paper, approved the final draft.

Alina Nalyvayko performed the experiments, authored or reviewed drafts of the paper, approved the final draft.

Yen Nguyen performed the experiments, authored or reviewed drafts of the paper, approved the final draft.

Benjamin Schwessinger conceived and designed the experiments, contributed reagents/materials/analysis tools, authored or reviewed drafts of the paper, approved the final draft.

Pamela C. Ronald conceived and designed the experiments, contributed reagents/materials/analysis tools, authored or reviewed drafts of the paper, approved the final draft.

The following information was supplied regarding data availability:

The data are represented in the Figures and Supplemental Files.

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
