# Peer review of "The rice XA21 ectodomain fused to the Arabidopsis EFR cytoplasmic domain confers resistance to Xanthomonas oryzae pv. oryzae"

_PeerJ, doi:10.7717/peerj.4456_

## Round 0.1 · original submission · Minor Revisions

· Academic Editor

Minor Revisions

Please address the comments of the reviewers, or explain why you cannot.

·

Basic reporting

The manuscript entitled “The rice XA21 ectodomain fused to the Arabidopsis EFR cytoplasmic domain confers resistance to Xanthomonas oryzae pv. oryzae" challenges the idea that the endodomain of plant plasma membrane localized pattern recognition receptors determine downstream immune response and the ectodomain is only responsible for pathogen molecule recognition. Using rice XA21 receptor (conferring resistance to bacterial pathogen Xoo virulent strain PXO99A) as an example, the authors propose the ectodomain of XA21 confers disease resistance and triggers immunity as far as a functional kinase present in endodomain. The authors introduced the background sufficiently, pointed out key major results from their own lab and other labs supporting the previous idea, also introduced hints against previous idea (chimera receptor with endodomain of XA21 and ectodomain of EFR which can recognize EF-TU from Xoo failed to confer resistance to Xoo strain PXO99A), which provided rationale for performing this work. The data of this ms are of quality in general, and well presented by the three figures.

Experimental design

The authors generated transgenic rice plants expressing chimera molecule composed of ectodomain of XA21 and endodomain of EFR (named XA21:EFR), T1 and T2 generation plants of multiple lines showed resistance to Xoo strain PXO99A based on lesion measurements after leaf inoculation. The application of sulfate peptide RaxX21-sY from Xoo, which can be recognized by Xa21, can trigger ROS production and some PR gene expression in XA21:EFR rice cells. The experimental design is pretty straightforward.

Validity of the findings

The conclusion that XA21 ectodomain confer resistance to Xoo is supported by the results here, in general. Statistically sound.

Additional comments

Just some material description can be more specific. 1. In the methods part, the authors should state which promoter was used in generating XA21:EFR transgenic rice, and whether it is the same promoter used to generate previous EFR:XA21 transgenic rice. Because promoter difference may also contribute to the difference in confer resistance to Xoo. 2. In introduction, the authors should introduce that Arabidopsis EFR ectodomain can recognize EF-TU from bacteria including Xoo. This will help readers to understand that previous report on chimera receptor with endodomain of XA21 and ectodomain of EFR didn’t increase resistance to Xoo, conflict with the idea endodomain of PPR determines immunity response. 3. In results part, the nature of LOC_Os02g36190, 275 LOC_Os06g37224, and LOC_Os11g42200 is vague. Whether they are PR genes or stress induced genes is not clear. A brief description of the nature of these genes can help readers to evaluate whether they can represent immunity response.

Reviewer 2 ·

Basic reporting

Good introduction of the chimera literature. Could maybe cut down the non-RD introduction, given it's not as relevant to a Xa21 vs EFR difference (they're both nonRD)

Experimental design

See point 5 in general comments

Validity of the findings

No major comments but see below

Additional comments

The manuscript Thomas et al. “The rice XA21 ectodomain fused to the Arabidopsis EFR cytoplasmic domain confers resistance to Xanthomonas oryzae pv. oryzae” describes experiments on transgenic rice lines that demonstrate an XA21 ectodomain is sufficient to confer RaxXsT response and Xoo resistance. While several additional experiments and controls would have be nice, the work is generally sound and contributes to the field of membrane receptor kinases and structural requirements for conferring novel resistance. There are some missing pieces of information, and several puzzling results are unexplained or uninterpreted – the points below should be addressed in a revision

1. In the discussion the authors interpret the resistance function of XA21:EFR relative to EFR:XA21 chimeras as a function of ectodomain interactors specific to Xa21. This is possible (and exciting), but an alternate explanation is that the strength of receptor activation in an Xoo-rice infection context is simply lower for activating an EFR ectodomain vs an XA21 ectodomain, as discussed by Schwessinger et al. 2015a. The authors should add this possibility to the discussion. Is there any previous data that addresses this in an Xoo infection context? E.g. similarity of early responses by Xoo (not synthetic elicitor) in EFR vs Xa21 transgenics would weigh against my alternative hypothesis. (Also possible that Xoo masks its elf epitope somehow?)

2. In Figure 4, EFR and EFR:Xa21 transgenics are shown to respond to RaxX. This strange result is not mentioned in the text or discussion. Is this actually elf18 treatment??

3. The resistance phenotype in Fig. 1, reduced lesion length, looks robust and the use of segregating lines is well explained. It would be very nice to also see reduced bacterial population and the use of a delta raxX strain as a negative control in these experiments. I think these would be required revision experiments in most journals, I defer to the editor here.

4. The scale of RLU in Fig 3 should be consistent across the graphs, and Fig. S4 should be included in the main text for easier comparison. Doing these steps would show that the ROS burst in the Xa21:EFR chimeric lines is much smaller than in native or EFR:Xa21 lines. It’s interesting that they still are resistant despite lower ability to be activated by synthetic elicitors (as seen with marker gene activation as well in Fig 4). The authors should note this general low magnitude of response in the results text.

5. Materials and methods is missing the protocol for bacterial infection. Also clarify that the elf18 sequence used is E.coli

Other comments:
Line 189 mentions using both RaxX21 and RaxX21-sY; I think it was only sY in this paper?

Lines 309-313: It’s unclear how differential kinase domain interactors are consistent with driving an ectodomain-specific effect. The data seem more consistent with these differential interactors not actually mattering for the actual resistance output, even if they affect ROS and other early events.

---

## Round 0.2 · accepted · Accept

· Academic Editor

Accept

Thanks for addressing the reviewer comments.